# What Comes after the Trial? An Observational Study of the Real-World Uptake of an E-Mental Health Intervention by General Practitioners to Reduce Depressive Symptoms in Their Patients

**DOI:** 10.3390/ijerph19106203

**Published:** 2022-05-19

**Authors:** Margrit Löbner, Janine Stein, Melanie Luppa, Markus Bleckwenn, Anja Mehnert-Theuerkauf, Steffi G. Riedel-Heller

**Affiliations:** 1Institute of Social Medicine, Occupational Health and Public Health, Medical Faculty, University of Leipzig, 04103 Leipzig, Germany; janine.stein@medizin.uni-leipzig.de (J.S.); melanie.luppa@medizin.uni-leipzig.de (M.L.); steffi.riedel-heller@medizin.uni-leipzig.de (S.G.R.-H.); 2Department of General Practice, Medical Faculty, University of Leipzig, 04103 Leipzig, Germany; markus.bleckwenn@medizin.uni-leipzig.de; 3Department of Medical Psychology and Medical Sociology, University Medical Center Leipzig, 04103 Leipzig, Germany; anja.mehnert@medizin.uni-leipzig.de

**Keywords:** depression, mental health, Internet intervention, general practitioner, primary care

## Abstract

Unguided and free e-mental health platforms can offer a viable treatment and self-help option for depression. This study aims to investigate, from a public health perspective, the real-world uptake, benefits, barriers, and implementation support needed by general practitioners (GPs). The study presents data from a spin-off GP survey conducted 2.5 years subsequent to a cluster-randomized trial. A total of *N =* 68 GPs (intervention group (IG) GPs = 38, control group (CG) GPs = 30) participated in the survey (response rate 62.4%). Data were collected via postal questionnaires. Overall, 66.2% of the GPs were female. The average age was 51.6 years (SD = 9.4), and 48.5% of the GPs indicated that they continued (IG) or started recommending (CG) the e-mental health intervention under real-world conditions beyond the trial. A number of benefits could be identified, such as ease of integration and strengthening patient activation in disease management. Future implementation support should include providing appealing informational materials and including explainer videos. Workshops, conferences, and professional journals were identified as suitable for dissemination. Social media approaches were less appealing. Measures should be taken to make it easier for health care professionals to use an intervention after the trial and to integrate it into everyday practice.

## 1. Introduction

Depression has been acknowledged as a major public health problem worldwide. According to the Global Burden of Disease report by the World Health Organization (WHO), depression is among the leading causes of disability [1]. E-mental health interventions, mostly relying on cognitive behavioral therapy (CBT), have shown their efficacy for a variety of mental health conditions [2,3,4,5], and especially for depression [6,7]. Electronic mental (e-mental) health involves digital interventions leveraging the Internet and related technologies, such as smartphone apps, websites, virtual reality-based interventions, or social media, to deliver mental health services, in order to promote and to improve mental health [8,9,10,11]. Even though effect sizes have been reported to be higher in guided interventions [12], unguided interventions have also been proven to be effective and are viable treatment and self-help options [13,14,15,16,17]. According to a new individual participant data (IPD) network meta-analysis, unguided Internet-based cognitive behavioral therapy (iCBT) is just as effective in improving depressive symptoms among individuals with mild and subthreshold depression as guided iCBT [17]. In addition, the superiority of guided interventions seems to diminish in the long term [17]. In the current German National Disease Management Guideline (S3), low-threshold psychosocial interventions such as online programs are an “open recommendation” for mild depressive episodes [18]. Due to the increasing evidence base, digital interventions are expected to receive a stronger recommendation grade for mild depression in the S3-Guideline, which is currently being updated. 

Most recently, however, evidence has emerged that the use of unguided e-mental health interventions may differ in real-world settings as compared to trial settings [19,20]. Recently published research suggests that the participation in a trial has a substantial impact on the user’s engagement in the e-mental health intervention [20]. Accordingly, adherence and completion rates of self-management interventions tend to be lower in real-world settings as compared to trial usage data [19,20]. Nevertheless, from a public health perspective, self-managed e-health interventions have the potential to reach a high number of people with little effort and minimal costs [21,22]. Because unguided iCBT has been effective in people with mild and subthreshold depression [17], general practitioners (GPs) or other related specialists could recommend unguided mental health platforms as adjunct treatment to other health care services. Additionally, health care professionals can increase the efficacy of e-health interventions simply by recommending them and asking follow-up questions about usage. It is likely that this sort of care by physicians with regard to recommending and following up on use of e-mental health interventions would have the same impact as trial procedures with ongoing assessments. Adherence and motivation to stick to the program might similarly be improved [23]. Yet, there is a current research gap with regard to investigating real-world usage from the perspective of health care professionals. General practitioners (GPs) play an especially important role as they are usually the first contact person for patients with depressive moods [24,25,26]. In addition, they hold an important gatekeeper function for further depression treatment options and specialist services [24,25]. The present study intends to investigate real-world data from GPs beyond trial participation. The preceding @ktiv trial was a cluster-randomized controlled trial (cRCT) regarding the effectiveness of a self-managed e-mental health intervention for improving depressive mood. The cRCT showed the intervention to be effective in reducing depressive symptoms in mild-to-moderately severe depressed patients in the short and long term compared to a control group [14]. The results on secondary outcomes also showed a long-term improvement in self-efficacy and quality of life. Gaining a better understanding about real-world uptake from the GP perspective could offer an important contribution to existing research. In addition, the present study aims to provide practical guidance for better implementation of an unguided iCBT program in primary care. Therefore, the following research questions are explored in the present study:(1)What was the rate of real-world uptake of the e-mental health intervention in GP practices after the trial?(2)How do GPs evaluate the real-world benefits of the e-mental health intervention?(3)What are the barriers to real-world uptake for GPs?(4)What future implementation support is needed to overcome barriers from the GP perspective?

## 2. Materials and Methods

### 2.1. Study Design and Sample

The present observational cross-sectional study presents quantitative data derived from a spin-off GP survey about 2.5 years subsequent to @ktiv trial. The study design and results of the preceding @ktiv trial have been described elsewhere [14,15,23,27]. Within the trial, *N =* 647 GP patients were recruited via *N =* 112 GP practices in central Germany. GPs received a letter when the online coaching platform was launched as freeware in Germany (1 January 2016). GPs did not know they were going to be surveyed 2.5 years after the trial as plans for the spin-off study did not yet exist at that time. No other reminders or incentives were given to implement the online coach into their regular treatment. Implementation rates, therefore, most likely reflect real-world conditions. The project was conducted in two steps. The first step involved a qualitative investigation (*N =* 3 GPs). Qualitative interviews were an important prerequisite to developing the GP questionnaire. In the second step, the quantitative GP survey followed. Hence, the remaining *N =* 109 physicians (*N =* 62 IG GPs; *N =* 47 GPs) were contacted and asked to participate. The subsequent GP survey was conducted from July 2018 to December 2018. Monetary incentives of EUR 80 were used to increase the response rate. Repeated reminders were mailed, following the Dillman model [28]. Overall, a sample of *N =* 68 GPs (IG GPs = 38, CG GPs = 30) were recruited. This corresponds to a response rate of 62.4%.

### 2.2. E-Mental Health Intervention

The @ktiv trial investigated the effectiveness of the German-language version of moodgym. Moodgym is a self-managed program based on theories and techniques of cognitive behavioral therapy. It consists of five interactive program modules that users can work through one after the other at the speed and intensity they choose. Interactive exercises, examples, and tests are integrated in each module. During the period of the @ktiv study, the German version of moodgym was only available to study participants in the intervention group. The German-language version has been available online free of charge since 2016. The further development of the program is guaranteed by ehub Health, a spin-off of the Australian National University (ANU), where moodgym was originally developed.

### 2.3. Assessment Instruments for Real-World Uptake and Implementation Experiences

The design of the survey instrument was based on the four core components of normalization process theory (NPT): coherence, cognitive participation, collective action, and reflexive monitoring [29]. The results of the qualitative GP interviews were incorporated into the questionnaire, for example, by adding specific questions. The evaluation of the questionnaire was carried out by means of a pre-test with *N =* 3 persons, who determined comprehensibility and clarity of the questionnaire. Subsequently, appropriate adjustments and modifications were made. The present study refers to the following questions integrated within the questionnaire:

#### 2.3.1. Socio-Demographic and Vocational Characteristics

Socio-demographic and vocational characteristics of the GP sample were assessed with regard to gender (female/male), age in years, specialist medical qualification (specialist in general medicine/general practitioner/specialist for internal medicine/other), additional qualifications in psychotherapy or psychosomatic medicine (yes/no).

#### 2.3.2. Real-World Uptake of the E-Mental Health Intervention

GPs were asked, “Did you recommend moodgym to patients after trial end of the @ktiv study?” (yes/no). In addition, “How many patients have you recommended use the program so far?” (1–5 patients/6–10 patients/11–15 patients/16–20 patients/more than 20 patients). 

#### 2.3.3. Real-World Benefit from the GP Perspective

GPs with ongoing use of the e-mental health intervention were asked to evaluate the benefit of the e-mental health intervention with regard to the following issues: (1) ease of integration as an additional treatment option in GP care, (2) as an additional tool when treating patients with depressive symptoms, (3) as a helpful treatment element, (4) as providing basic theoretical knowledge about depressive illness, and (5) as a tool to strengthen the personal responsibility of the patient. Answering options were: exactly true/rather true/ rather not true/ not true at all).

#### 2.3.4. Barriers for Real-World Uptake from the GP Perspective

GPs were asked to evaluate barriers to real-world uptake of the e-mental health intervention with the following question: “If you have not used moodgym, what were the reasons?” Multiple response options were possible: (Simply forgot/Lack of interest from patients/Negative feedback from patients/Concerns about whether patients will be able to manage the use on their own /Anonymity of the Internet/Doubts about the effectiveness of the program/Missing informational materials to hand out/Lack of time during a patient consultation/Lack of preparation time before a patient consultation/Stress due to high patient volume/Uncertain who the program is suitable for/Insufficient knowledge about the program/Privacy concerns).

#### 2.3.5. Future Implementation Support Needed

GPs who continued to use the e-mental health intervention were asked the following questions to evaluate future implementation support needed to overcome barriers: (1) Which of the following measures do you think are necessary, from a GP perspective, in order to use moodgym in primary care? (An online training seminar for the program/A video of the various modules (5–10 min)/A video example of the doctor–patient conversation during a consultation/Informational materials (e.g., flyers) to pass on to patients/The availability of a telephone consultation hotline for physicians/Consultancy support by psychiatrists/psychotherapists via video conference); (2) How could the awareness for moodgym be raised among primary care physicians? (Treatment guideline (S3 guideline)/Publications in professional journals/Professional societies/Congresses/conferences/Training courses/Workshops/Internet/Health insurance companies/Social media (e.g., Facebook, Twitter)/Informational material by postal mail); (3) Would you like to involve your practice staff in the use of moodgym in your practice? (yes/no).

### 2.4. Statistical Analyses

All statistical analyses were performed using the Statistical Package for the Social Sciences 25.0 for Windows (SPSS Inc., Chicago, IL, USA). Descriptive data are presented as means with standard deviations (SD) or absolute frequencies and percentages within the result section. Inferential statistics (Chi-square tests) were used to investigate differences between IG and CG GPs. All analyses are based on a level of significance with a *p*-value below 0.05. 

## 3. Results

### 3.1. Socio-Demographic and Vocational Characteristics of the Study Sample

Table 1 shows an overview of the socio-demographic and vocational characteristics of the participating GPs (in total, for IG and CG). A total of 66.2% of the GPs were female. The average age was 51.6 years (SD = 9.4). More than half of the sample (67.6%) stated that they had a specialist qualification for general medicine, and the next most common specialty was internal medicine (29.4%), with 69.1% stating that they had an additional qualification in psychotherapy or psychosomatic medicine. There were no significant differences between IG and CG GPs.

### 3.2. Real-World Uptake of the E-Mental Health Intervention 

About half (*n* = 33, 48.5%) of the *N =* 68 GPs indicated that they carried on recommending moodgym (IG) or started recommending it (CG) under real-world conditions. That is, 44.7% (*n* = 17) of the IG GPs and 53.3% (*n =* 16) of the CG GPs recommended the program to patients. From *N =* 32 of the participating GPs, information was available with regard to the number of patients to whom the program was recommended to after the trial ended. Hence, 12.5% (*n* = 4) recommended the program to 1–5 patients, 40.6% (*n =* 13) recommended the program to 6–10 patients, 21.9% (*n* = 7) of the GPs recommended the program to 11–15 patients, and 15.6% (*n* = 5) of the GPs recommended the program to 16–20 patients. A total of 9.4% (*n* = 3) GPs recommended the program to more than 20 patients.

### 3.3. Real-World Benefit from GP’s Perspectives

Table 2 shows the evaluation of the real-world benefit from the GP perspective regarding the use of the e-mental health intervention after the trial end. The majority of GPs agreed that integration in GP care was easy (“rather agree” (62.5%) or “strongly agree” (31.5%)). Most GPs evaluated the intervention to be an additional tool they could use when treating patients with depressive symptoms (“exactly true” (34.4%), “rather true” (59.4%)) and found it helpful to be able to actively recommend the intervention (“exactly true” (40.6%), “rather true” (56.3%)). The majority of GPs agreed that the intervention provided basic knowledge about depression (“exactly true” (21.9%), “rather true” (62.5%)) and strengthened patient activation in the management of disease (“exactly true” (53.1%), “rather true” (40.6%)). IG GPs showed significantly stronger agreement than CG GPs with regard to ease of integration (*p* < 0.01) and strengthening patient activation in the management of disease (*p* < 0.05).

### 3.4. Barriers for Real-World Uptake from the GP Perspective

GPs who did not use of the e-mental health intervention beyond the trial were asked to describe their individual barriers to real-world uptake. Results are shown in Table 3. Most frequently, GPs endorsed the following barriers: simply forgot (85.2%, *n =* 23), followed by stress due to high patient volume (77.8%, *n =* 21), and not having informational hand-outs (76.9%, *n =* 20). No significant differences between IG or CG GPs were observed.

### 3.5. Future Implementation Support Needed

GPs with ongoing use of the e-mental health intervention were asked for their opinion on supporting measures for implementation. Results are shown in Table 4. Most frequently, informational materials (e.g., flyers) for passing on to the patients was mentioned here (96.9%, *n =* 31), followed by a 5 to 10 min video that leads the individual through program modules (80%, *n =* 24). More than two-thirds (63.3%, *n =* 19) stated that they would not consider consultancy support by psychiatrists/psychotherapists via video conference as necessary. No significant differences between IG or CG GPs were observed.

Table 5 shows the possibilities for increasing the level of awareness for GPs. A proportion of 90.0% indicated training courses and workshops as useful tools to raise the level of awareness among GPs. The same applies for congresses and conferences (86.2%). In total, 83.3% of the GPs mentioned publications in professional journals. Social media (e.g., Facebook, Twitter) were mentioned less often (13.8%) by both IG and CG GPs. Nevertheless, CG GPs were significantly more open-minded with regard to this option.

GPs with ongoing use of the e-mental health intervention indicated, by a propor-tion of 77.4% (*N* = 24), that they would like to involve their practice staff with regard to the use of the e-mental health intervention. No significant differences in IG and CG patients were observed (Chi^2^ = 0.278; *p* = 0.598).

## 4. Discussion

The current study delivers data for the first time on real-world uptake of an e-mental health intervention for depressive symptoms from the perspective of primary care physicians. Participation in this study stemmed from participation in a previous study after which about half of the GPs reported to have carried on recommending the intervention (IG) or to have started recommending it (CG) under real-world conditions. Various real-world benefits were indicated by those GPs using the intervention under real-world conditions, such as an ease of integration, usefulness, patient satisfaction, ability to recommend an additional tool, and activating patient self-management of disease. 

Still, study findings imply that a considerable number of GPs did not carry on (IG) or start (CG) using the e-mental health intervention beyond the trial. This is an important research finding about real-world uptake that has received little attention in other trials so far. The present study offers for the first time insight into the possible barriers from a GP point of view. The most relevant hindrances to the implementation of the e-mental health intervention were simply forgetting it and stress due to high patient volume. Patient information materials, such as flyers and posters, could be used in the waiting room to make interested patients aware of the possibility of e-mental health platforms and to talk to their GP about it. However, another prominent barrier was the lack of informational materials to hand out, which is an important hint for implementation strategies after trial. Developers of e-mental health interventions should be aware of this need. E-Mental health projects should plan to develop specific information material to achieve successful implementation at the end of a trial. On the other hand, intervention-related issues such as privacy concerns, doubts about the effectiveness of the program, or the anonymity of the Internet were rarely stated as a barrier. Similarly, patient-related issues such as negative feedback or lack of interest from patients did not seem to play a major role. Especially with the backdrop of the most recently published research that indicates that patient use of unguided e-mental health interventions may differ in real-world settings as compared to trial settings [19,20], health care professionals’ informed recommendations and active inquiries could make a valuable contribution here. Facilitating the use of unguided e-mental health interventions for health-care professionals, such as GPs, might have a favorable impact on uptake, adherence, and even effectiveness in patients with depressive symptoms. In addition, even relatively brief use of an e-mental health intervention might have a significant population impact by conveying the message that depression is a common health problem and that there are ways of treating it [19].

No significant differences in terms of real-world usage were shown between IG GPs and CG GPs. Similarly, both groups showed similar responses with regard to perceived benefits, barriers, and implementation support. Only minor differences were found, such as the control group GPs being more likely to favor social media as a source of information to increase the level of awareness for e-mental health interventions. The present findings indicate that the allocation group of recruiting health-care professionals in cRCTs has no specific influence on the recommendation behavior after the trial. Future studies are needed to verify these results.

Interestingly, even though other online coaches for depression were available in Germany after @ktiv trial ended, most of the GPs in the spin-off survey (94%) stated that they were not aware of any other programs in this field. Consequently, GPs were not using any alternative depression online coach within these 2.5 years. This finding also implies that there might be still a lack of information about e-mental health treatment options with regard to primary care physicians. In Germany, the legal framework for the prescription of digital health applications has just recently been established with the Digitale-Versorgungs-Gesetz (DVG), which came into force in 2019 [30]. Since 2020, costs for digital health applications listed in a so-called DiGA directory can be billed directly to health insurance companies. Still, measures need to be taken to explain the benefits and advantages of e-mental health interventions and to strengthen knowledge about their use in patient care. The present study makes some suggestions with regard to GPs. The heterogeneity of results in the spin-off survey points toward the application of broadly diversified measures, since one measure may not fit all. Accordingly, the present study shows that the majority of GPs favors certain implementation aids (e.g., information material 96.9%, application videos 80.0%). Consequently, future implementation support should include appealing informational materials and explainer videos. Other implementation aids are favored less often, but still by a considerable number of GPs (e.g., online training seminars 50.0%). The results suggest that there might be the need to offer a range of implementation support measures, leaving it to the health care professional to choose an appropriate measure. 

In the present study, the majority of GPs affirmed the listed strategies to increase the level of awareness. The dissemination of information should be considered via workshops, conferences, and professional journals. In addition, information via treatment guidelines and professional societies seem to play an important role. Another strategy could involve the distribution of information material via Internet, by postal mail, or via health insurance companies. Contact via social media was less appealing for GPs. 

Implementation studies that develop and evaluate the effectiveness of different implementation measures are urgently needed. Future studies should not only focus on potential users of the e-mental health intervention, but also on health care professionals and their staffs. Patients and health care professionals should be involved in the development of implementation strategies. Mixed-method studies are therefore strongly recommended. Longitudinal observational studies could be useful to track the implementation success of the developed measures. In addition, measuring an increase in the number of users of e-mental health interventions could also be an appropriate way to measure the effect of implementation measures. 

In addition, the treatment preferences of patients should be considered by their treating physicians [31,32]. Moreover, a lack of teaching about how to use e-mental health interventions has been identified in previous research as an important barrier for successful implementation [33,34,35]. Incorporating e-mental health curricula within the academic training of health-care professionals, such as GPs, psychiatrists, or psychotherapists could be a valuable solution. 

There are several strengths of the study. The present study closes a research gap by presenting data on real-world uptake of an e-health intervention for improving depressing symptoms in patient care beyond the trial. For the first time, barriers of real-world uptake could be assessed from the GP perspective. In contrast to previous studies focusing on user uptake only, the present work refers to uptake from the perspective of health care professionals, namely GPs, which has never before been investigated. In addition, the assessment instrument used in the quantitative part of the study was adapted to important issues indicated by *N =* 3 qualitative GP interviews. Therefore, significant topics for this target group could assumingly be captured. However, the present study is not without limitations. Data refer to a single trial and to a sole intervention. Monetary incentives were used to increase return rates of the survey. For this reason, a possible response bias cannot completely be ruled out and responses may not fully reflect real-world usage. Generalizability of the results may therefore be limited. More research data are needed, and future studies should take the investigation of real-world usage data from the health care professionals’ perspective into account. Furthermore, the response rate of GPs participating in the add-on survey after trial was only 62.4%. Low response rates in physician surveys are a common problem in research and often due to time constraints [36]. Reasons for non-participation could not be assessed in the present study. Percentages of real-world uptake within patient care should therefore be interpreted with caution and might be lower than reported. In addition, no validated instrument was available to assess real-world uptake, real-world benefits, barriers, and implementation support from the GP’s perspective. For this reason, the study was following a sequential qualitative-quantitative research design, also called exploratory sequential mixed-methods design [37]. Accordingly, qualitative GP interviews were used to develop survey variables and survey items for the quantitative study part. In addition, a pretest (*N =* 3) was used to identify and to modify any ambiguities. Nevertheless, the validation of this survey instrument should be a next step to enable other studies to collect valid data on real-world usage.

## 5. Conclusions

Unguided iCBT interventions have been proven to effectively improve depressive symptoms. Especially with regard to mild forms of depression or subthreshold conditions, these interventions could be a useful and economical tool as add-on to patient care. GPs do not only hold a gate-keeper function with regard to specialist service, moreover, patients trust their opinion and recommendations. Thus, GPs’ active support of unguided e-mental health interventions may enhance uptake, adherence and due to close dose-response relationships even significantly impact their effectiveness. Measures should be taken to make it easier for health care professionals to use an intervention after the trial and to integrate it into everyday practice. Offering appealing informational materials and explainer videos would help to encourage participation. More research is needed to investigate real-world uptake and implementation approaches for e-mental health interventions from both patients’ and professionals’ perspectives.

## Figures and Tables

**Table 1 ijerph-19-06203-t001:** Socio-demographic and vocational characteristics.

Variables		Total (*n* = 68)	IG (*n* = 38)	CG (*n* = 30)	Chi^2^/t	*p*
gender, *n* (%)	female	45 (66.2)	26 (68.4)	19 (63.3)	0.194	0.660
	male	23 (33.8)	12 (31.6)	11 (36.7)		
age in years, mean (SD)		51.6 (9.4)	51.7 (9.6)	51.4 (9,3)	0.160	0.874
specialist medical qualification, *n* (%) *	specialist in general medicine	46 (67.6)	24 (63.2)	22 (73.3)	0.793	0.373
	general practitioner	5 (7.4)	2 (5.3)	3 (10.0)	0.552	0.457
	specialist for internal medicine	20 (29.4)	12 (31.6)	8 (26.7)	0.195	0.659
	other	9 (13.2)	3 (7.9)	6 (20.0)	2.139	0.144
additional qualifications in psychotherapy or psychosomatic medicine, *n* (%)	yes	47 (69.1)	29 (76.3)	18 (60.0)	2.091	0.148
	no	21 (30.9)	9 (23.7)	12 (40.0)		

Notes. IG = intervention group; CG = control group; SD = standard deviation; * multiple answers were possible.

**Table 2 ijerph-19-06203-t002:** Real-world benefit from the GP perspective ^1^.

Variables		All GPs *n* (%)	IG GPs *n* (%)	CG GPs *n* (%)	Chi^2^	*p*
Moodgym can be easily integrated as an additional treatment option in GP care.	strongly agree	10 (31.3)	9 (56.3)	1 (6.3)	10.200	**0.006 ****
	rather agree	20 (62.5)	7 (43.8)	13 (81.3)		
	rather disagree	2 (6.3)	0 (0)	2 (12.5)		
	strongly disagree	0 (0)	0 (0)	0 (0)		
Moodgym offers me an additional “tool” that I can use when treating patients with depressive symptoms.	exactly true	11 (34.4)	8 (50.0)	3 (18.8)	3.589	0.166
	rather true	19 (59.4)	7 (43.8)	12 (75.0)		
	rather not true	2 (6.3)	1 (6.3)	1 (6.3)		
	not true at all	0 (0)	0 (0)	0 (0)		
I find it helpful to be able to actively recommend moodgym as a treatment element.	exactly true	13 (40.6)	9 (56.3)	4 (25.0)	3.812	0.149
	rather true	18 (56.3)	7 (43.8)	11 (68.8)		
	rather not true	1 (3.1)	0 (0)	1 (6.3)		
	not true at all	0 (0)	0 (0)	0 (0)		
The use of moodgym enables me to provide my patients with basic theoretical knowledge about their depressive illness.	exactly true	7 (21.9)	5 (31.3)	2 (12.5)	2.486	0.478
	rather true	20 (62.5)	9 (56.3)	11 (68.8)		
	rather not true	4 (12.5)	2 (12.5)	2 (12.5)		
	not true at all	1 (3.1)	0 (0)	1 (6.3)		
Moodgym strengthens the personal responsibility of the patient to actively contribute to the management of his depressive symptoms.	exactly true	17 (53.1)	12 (75.0)	5 (31.3)	6.805	**0.033 ***
	rather true	13 (40.6)	4 (25.0)	9 (56.3)		
	rather not true	2 (6.3)	0 (0)	2 (12.5)		
	not true at all	0 (0)	0 (0)	0 (0)		

Notes. ^1^ Sample size refers to the subsample of *N =* 33 GPs who reported real-world uptake of the e-mental health intervention. Information on real-world benefit was only available from *N =* 32 GPs. IG = Intervention group; CG = control group; * *p* < 0.05, ** *p* < 0.01.

**Table 3 ijerph-19-06203-t003:** Barriers for real-world uptake from GP’s perspectives ^1^.

Variables		All GPs *n* (%)	IG GPs *n* (%)	CG GPs *n* (%)	Chi^2^	*p*
Simply forgot (*n* = 27)	yes	23 (85.2)	10 (76.9)	13 (92.9)	1.356	0.244
	no	4 (14.8)	3 (23.1)	1 (7.1)		
Lack of interest from patients (*n* = 26)	yes	10 (38.5)	5 (41.7)	5 (35.7)	0.097	0.756
	no	16 (61.5)	7 (58.3)	9 (64.3)		
Negative feedback from patients (*n* = 25)	yes	2 (8.0)	2 (16.7)	0 (0)	2.355	0.125
	no	23 (92.0)	10 (83.3)	13 (100)		
Concerns about whether patients will be able to manage the use on their own (*n* = 26)	yes	9 (34.6)	6 (46.2)	3 (23.1)	1.529	0.216
	no	17 (65.4)	7 (53.8)	10 (76.9)		
Anonymity of the Internet (*n* = 26)	yes	2 (7.7)	1 (7.7)	1 (7.7)	0.000	1.000
	no	24 (92.3)	12 (92.3)	12 (92.3)		
Doubts about the effectiveness of the program (*n* = 25)	yes	3 (12.0)	1 (8.3)	2 (15.4)	0.294	0.588
	no	22 (88.0)	11 (91.7)	11 (84.6)		
Missing informational material to hand out (*n* = 26)	yes	20 (76.9)	8 (66.7)	12 (85.7)	1.321	0.250
	no	6 (23.1)	4 (33.3)	2 (14.3)		
Lack of time during a patient consultation (*n* = 25)	yes	12 (48.0)	6 (50.0)	6 (46.2)	0.037	0.848
	no	13 (52.0)	6 (50.0)	7 (53.8)		
Lack of preparation time before patient consultation (*n* = 25)	yes	9 (36.0)	3 (25.0)	6 (46.2)	1.212	0.271
	no	16 (64.0)	9 (75.0)	7 (53.8)		
Stress due to high patient volume (*n* = 27)	yes	21 (77.8)	9 (69.2)	12 (85.7)	1.060	0.303
	no	6 (22.2)	4 (30.8)	2 (14.3)		
Uncertain who the program is suitable for (*n* = 27)	yes	10 (37.0)	5 (38.5)	5 (35.7)	0.022	0.883
	no	17 (63.0)	8 (61.5)	9 (64.3)		
Insufficient knowledge about the program (*n* = 26)	yes	16 (61.5)	5 (41.7)	11 (78.6)	3.718	0.054
	no	10 (38.5)	7 (58.3)	3 (21.4)		
Privacy concerns (*n* = 25)	yes	4 (16.0)	2 (16.7)	2 (15.4)	0.008	0.930
	no	21 (84.0)	10 (83.3)	11 (84.6)		

Notes. ^1^ Sample size refers to the subsample of *N =* 35 GPs who reported no use of the e-mental health intervention beyond trial. IG = intervention group; CG = control group.

**Table 4 ijerph-19-06203-t004:** Necessary implementation measures ^1^.

Variables		All GPs *n* (%)	IG GPs *n* (%)	CG GPs *n* (%)	Chi^2^	*p*
Online training seminar for the program (*n* = 29)	yes	14 (48,3)	7 (50.0)	7 (46.7)	0.032	0.858
	no	15 (51.7)	7 (50.0)	8 (53.3)		
Video that demonstrates the modules (5–10 min) (*n* = 30)	yes	24 (80.0)	12 (80.0)	12 (80.0)	0.000	1.000
	no	6 (20.0)	3 (20.0)	3 (20.0)		
Video example of the doctor–patient conversation during a consultation (*n* = 29)	yes	12 (41.4)	7 (46.7)	5 (35.7)	0.358	0.550
	no	17 (58.6)	8 (53.3)	9 (64.3)		
Informational materials (e.g., flyers) to pass on to the patients (*n* = 32)	yes	31 (96.9)	16 (100)	15 (93.8)	1.032	0.310
	no	1 (3.1)	0 (0)	1 (6.3)		
Availability of a telephone consultation hotline for physicians (*n* = 30)	yes	14 (46.7)	6 (40.0)	8 (53.3)	0.536	0.464
	no	16 (53.3)	9 (60.0)	7 (46.7)		
Consultancy support by psychiatrists/psychotherapists via video conference (*n* = 30)	yes	11 (36.7)	8 (50.0)	3 (21.4)	2.625	0.105
	no	19 (63.3)	8 (50.0)	11 (78.6)		

Notes. ^1^ Sample size refers to the subsample of *N =* 33 GPs, who reported real-world uptake of the e-mental health intervention. IG = intervention group; CG = control group.

**Table 5 ijerph-19-06203-t005:** Possibilities to increase the level of awareness ^1^.

Variables		All GPs n (%)	IG GPs n (%)	CG GPs n (%)	Chi^2^	*p*
Treatment guideline (S3 guideline) (*n =* 30)	yes	23 (76.7)	13 (81.3)	10 (71.4)	0.403	0.526
	no	7 (23.3)	3 (18.8)	4 (28.6)		
Publications in professional journals (*n =* 30)	yes	25 (83.3)	12 (75.0)	13 (92.9)	1.714	0.190
	no	5 (16.7)	4 (25.0)	1 (7.1)		
Professional societies (*n =* 28)	yes	19 (67.9)	11 (68.8)	8 (66.7)	0.014	0.907
	no	9 (32.1)	5 (31.3)	4 (33.3)		
Congresses/conferences (*n =* 29)	yes	25 (86.2)	15 (93.8)	10 (76.9)	1.708	0.191
	no	4 (13.8)	1 (6.3)	3 (23.1)		
Training courses/workshops (*n =* 30)	yes	27 (90.0)	14 (87.5)	13 (92.9)	0.238	0.626
	no	3 (10.0)	2 (12.5)	1 (7.1)		
Internet (*n =* 27)	yes	15 (55.6)	9 (64.3)	6 (46.2)	0.898	0.343
	no	12 (44.4)	5 (35.7)	7 (53.8)		
Health insurance companies (*n =* 30)	yes	22 (73.3)	12 (75.0)	10 (71.4)	0.049	0.825
	no	8 (26.7)	4 (25.0)	4 (28.6)		
Social media (e.g., Facebook, Twitter) (*n =* 29)	yes	4 (13.8)	0 (0)	4 (30.8)	5.711	**0.017 ***
	no	25 (86.2)	16 (100)	9 (69.2)		
Informational material by postal mail (*n =* 30)	yes	19 (63.3)	12 (75.0)	7 (50.0)	2.010	0.156
	no	11 (36.7)	4 (25.0)	7 (50.0)		

Notes. ^1^ Sample size refers to the subsample of *N =* 33 GPs who reported real-world uptake of the E-mental health intervention. IG = intervention group; CG = control group; * *p* < 0.05.

## Data Availability

The datasets generated and analyzed during the current study are not publicly available due ethical restrictions and participant confidentiality but are available from the corresponding author on reasonable request. Aggregated data are provided in the paper’s tables.

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
