# Peer review of "What Comes after the Trial? An Observational Study of the Real-World Uptake of an E-Mental Health Intervention by General Practitioners to Reduce Depressive Symptoms in Their Patients"

_ijerph, 2022, doi:10.3390/ijerph19106203_

Round 1

Reviewer 1 Report

The topic of the paper is particularly interesting and topical, as it deals with the effect of apps for mental health in the treatment of psychiatric pathologies, in this case of depression, demonstrating their usefulness. The article is well written and understandable, the method used is thorough and precise. I recommend publishing it as it is, with the exception of a few small suggestions to the authors:

1) I recommend to include in the introduction a brief explanation and / or a definition of what is meant by "E-mental intervention" because the topic may not be familiar or known to all readers.

2) improve the provision of the tables that are partly overlapping with the text or slightly detached, making the reading a little confused.

Author Response

Thank you very much for your review and your helpful remarks. Please find our answers and amendments to your comments below:

1. We now provide an explanation for e-mental health interventions within the introduction section:

„Electronic mental (E-mental) health involves digital interventions leveraging the Internet and related technologies such as smartphone apps, websites, virtual reality-based interventions or social media to deliver mental health services, to promote and to improve mental health [8-11].“ (line 44 - line 47, tracked changes modus)

2. We have revised the tables according to your advice and hope that this has improved readability. (please see table 1-5)

Reviewer 2 Report

I am particularly interested in public health approaches to mental health intervention and supportive measures, as well as adjunctive efforts for general practitioners. I am grateful for your work and think that it is meaningful, relevant, and potentially significant in the many efforts to improve mental health outcomes. A few recommendations.

Many portions of the English was difficult to follow. (e.g. is even similar effective; Especially in view of unguided iCBT being comparably effective in people with mild and subthreshold depression [13], their use as an adjunct to other health care services, such as general practitioner 61 (GP) or specialist care may be beneficial for this specific target group; etc.)

Provider incentive may be a study limitation due to influence on responses and may not fully reflect real world usage. Clarification of study limitations is needed.

Methodologically, reliability and validity of study measures need to be stated. One was stated as validated by N=3, which is not a large enough sample to validate.

Few significant results were found in the study. Further study with validated measures may yield more significant findings.

Line 44 - please define IPD and iCBT

Line 72 -  clarification of @ktiv as previous study needed

Author Response

Thank you very much for reviewing our manuscript and for your helpful comments. Please find our answers and amendments below:

1. An English native speaker has revised the manuscript according to your advice and we hope that English language has now imroved. Please follow the tracked changes within the manuscript.

2. We thank you for this important comment. We now dicuss this as a limitation within the discussion section:

„Monetary incentives were used to increase return rates of the survey. For this reason, a possible response bias cannot completely be ruled out and responses may not fully reflect real world usage.“ (line 390 – line 392, tracked changes modus)

3. Thank you for this valuable remark. Unfortunately, at this point no validated instruments exist with regard to the research questions of our study. For this reason, the study was following a sequential qualitative-quantitative research design, also called exploratory sequential mixed methods design, which is an appropriate procedure in this case (Creswell, J. W., & Plano Clark, V. L. (2018). Designing and conducting mixed methods research. Thousand Oaks, CA: SAGE.). Our study started with an exploratory, qualitative phase and moved sequentially to a quantitative phase. The qualitative phase concluded with analysis producing codes or more conceptual themes. The results of this analysis was used to direct the next, quantitative phase by developing a survey for quantitative data collection. The qualitative research part included three qualitative interviews as an important prerequisite for the development of the GP questionnaire. The development of the qualitative interview guide was based on the Normalization Process Theory (NPT) according to May and Finch (2009) within the framework of a "qualitative research workshop". For the qualitative data collection, three general practitioners were interviewed by telephone until saturation with regard to the study content was reached. The interview content was tape-recorded. Subsequently, they were completely transcribed and evaluated by a combination of inductive and deductive qualitative content analysis according to Mayring using MAXQDA 11 software. The results were used to develop survey variables and survey items fort he quantitative study part. The purpose of the pretest (N=3) was to assess the usability of the questionnaire in order to identify and, if necessary, modify any ambiguities. Nevertheless, the validation of this survey instrument should be a next step and we added this within our discussion section. In addition, we go more into detail with regard to the exploratory sequential mixed methods design within the discussion section:

„In addition, no validated instrument was available to assess real-world uptake, real-world benefits, barriers and implementation support from GP´s perspective. For this reason, the study was following a sequential qualitative-quantitative research design, also called exploratory sequential mixed methods design [38]. Accordingly, qualitative GP interviews were used to develop survey variables and survey items for the quantitative study part. In addition, a pretest (N=3) was used to identify and to modify any ambiguities. Nevertheless, the validation of this survey instrument should be a next step to enable other studies to collect valid data on real world usage.“ (line 399 - line 406, tracked changes modus)

4. Thank you very much for this comment. The exploratory research questions of our study aimed at describing rather than comparing the real-world uptake, real-world benefits, barriers and needed future implementation support from GP´s perspective. Results are presented for the whole sample, but also separately for intervention group GPs and control group GPs. The comparison between both allocation group GPs, was displayed, because we hypothesized that group allocation might have had an impact on realworld uptake of the e-mental health intervention. More precisely, we hypothesized that control group GPs, who did not recommend the intervention during the trial, might have had a lower affinity for recommending the intervention because they had less experience with it. Therefore it was a very interesting finding for us that only few results differed between both groups. We now add a paragraph within the discussion section that goes more into detail about this issue. Please note, that the survey was not developed to distinguish between both allocation groups.

„No significant differences in terms of real world usage were shown between IG GPs and CG GPs. Likewise, both groups showed similar responses with regard to perceived benefits, barriers and implementation support. Only minor differences were found, such as the control group GPs being more likely to favor social media as a source of information to increase the level of awareness for e-mental health interventions. The present findings indicate that the allocation group of recruiting health care professionals in cRCTs has no specific influence on the recommendation behavior after the trial. Future studies are needed to verify these results.“ (line 330 - line 337, tracked changes modus)

5. We now provide a definition for IPD and iCBT.

„According to a new individual participant data (IPD) network meta-analyses unguided internet-based cognitive behavioral therapy (iCBT) is even similar effective to …“ (line 50 - line 51)

6. Thank you for this important comment. The word @ktiv is not an acronym. It stands for the German word „aktiv“ (English: active). The @ sign is meant to allude to the digital intervention. In the context of the study the word @ktiv stands for „actively fighting depression“. Please note, that we were asked to shorten the method section with regard to the trial description to avoid overlaps with the introduction. Instead we made an amendment with regard to other publications (secondary analyses) of the @ktiv trial. We hope these changes make it more easy to follow the study procedure.

„The preceding @ktiv trial was a cluster-randomized controlled trial (cRCT) regarding the effectiveness of a self-managed E-mental health intervention for improving depressive mood. The cRCT showed the intervention to be effective in reducing depressive symptoms in mild to moderately severe depressed patients in the short and long-term compared to a control group [14]. The results on secondary outcomes also showed a long-term improvement in self-efficacy and quality of life.“ (line 86 - line 92, tracked changes modus)

„The present observational cross-sectional study presents quantitative data deriving from a spin-off GP survey about 2.5 years subsequent to @ktiv trial. The study design and results of the preceding @ktiv trial have been described elsewhere [14,15,27,28]. Within the trial N=647 GP patients were recruited via N=112 GP practices in central Germany. GPs received a letter when the online coaching platform was launched as freeware in Germany (January 1, 2016). …“  (line 110 - line 123, tracked changes modus)

Reviewer 3 Report

This original research is of great interest to the practice of helping people with symptoms of depression.

The article contains all the necessary sections, a clear structure and logic of presentation of the material.

The "Introduction" section presents an analysis of the research problem, its importance and relevance.
The "Methods and design of the study" are described in detail.
The following is an analysis of the data and the results of the study, the conclusions are correctly formulated.
It is important to emphasize that the authors paid attention to the discussion of the results obtained. Such, the discussion section shows how the results of this study have implications for the development and possible barriers to E-mental health interventions.

It is important that the authors also note the importance of such issues as confidentiality, the availability of feedback, the attitudes of patients to using "of unguided E-mental health interventions".

The article can be recommended for publication.

Author Response

Thank you very much for your review. We appreciate your valuable comments.

Reviewer 4 Report

Depression is an issue socially important, which is why the lines of research that are generated to deal with are indispensable. Therefore, the paper is important in the dissemination of effective interventions for people who deal with depression. However, two aspects are considered worthy of attention:

1. Review the method because turns out to be repetitive regarding the introduction, and at times the goal of the work gets lost.  2. In the discussion, it would be important to make a deep analysis regarding the limitations of the study and proposals for future investigations. On the other hand to the results, basically: a) on the barriers and how it is intended to be solved, b) necessary implementation measures for the majority of the reagents almost 50% say yes and the other 50% say no, c) possibilities to increase the level of awareness, on all of the reagents the majority say yes. 

Author Response

Thank you very much for your review and your helpful remarks. Please find our answers and amendments to your comments below:

1. Thank you very much for this valuable comment. We thouroghly revised our methods section to avoid the repetitive phrases with regard to the introduction.

„The present observational cross-sectional study presents quantitative data deriving from a spin-off GP survey about 2.5 years subsequent to @ktiv trial. The study design and results of the preceding @ktiv trial have been described elsewhere [14,15,27,28]. Within the trial N=647 GP patients were recruited via N=112 GP practices in central Germany. GPs received a letter when the online coaching platform was launched as freeware in Germany (January 1, 2016). GPs did not know they were going to be surveyed 2.5 years after the trial as plans for the spin-off study did not exist at that time….“ (line 110 - line 127, tracked changes modus)

2. This is an important comment and revised the discussion section of the manuscript accordingly:

  • We now go more into detail with regard to the limitations of our study:

„However, the present study is not without limitations. Data are referring to a single trial and to a sole intervention. Monetary incentives were used to increase return rates of the survey. For this reason, a possible response bias cannot completely be ruled out and re-sponses may not fully reflect real world usage. Generalizability of results may, therefore, be limited. More research data are needed, and future studies should take the investigation of real-world usage data from the health care professionals’ perspective into account. Furthermore, the response rate of GPs participating in the add-on survey after trial was only 62.4%. Low response rates in physician surveys a common problem in research and often due to time constraints [3137]. Reasons for non-participation could not be assessed in the present study. Percentages of real-world uptake within patient care should therefore be interpreted with caution and might be lower than reported. In addition, no validated instrument was available to assess real-world uptake, real-world benefits, barriers and implementation support from GP´s perspective. For this reason, the study was following a sequential qualitative-quantitative research design, also called exploratory sequential mixed methods design [38]. Accordingly, qualitative GP interviews were used to develop survey variables and survey items for the quantitative study part. In addition, a pretest (N=3) was used to identify and to modify any ambiguities. Nevertheless, the validation of this survey instrument should be a next step to enable other studies to collect valid data on real world usage.“ (line 388 - line 406, tracked changes modus)

  • We added more detailled information with regard to proposals for future investigations:

„Implementation studies that develop and evaluate the effectiveness of different implementation measures are urgently needed. Future studies should not only focus on potential users of the e-mental health intervention, but also on health care professionals and their staff. Patients and health care professionals should be involved in the development of implementation strategies. Mixed method studies are therefore strongly recommended. Longitudinal observational studies could be useful to track the implementation success of the developed measures. In addition, measuring an increase in the number of users of E-Mental health interventions could also be an appropriate way to measure the effect of implementation measures.“ (line 365 - line 373, tracked changes modus)

  • We now adress more detailed the important points you made (a, b, c):

a) on the barriers and how it is intended to be solved:

„The present study offers for the first time insight on possible barriers from a GP point of view. The most relevant hinderances to the implementation of the E-mental health intervention were simply forgetting it and stress due to high patient volume. Patient information materials, such as flyers and posters could be used in the waiting room to make interested patients aware of the possibility of e-mental health platforms and to talk to their GP about it. However, another prominent barrier was the lack of informational materials to hand out, which is an important hint for implementation strategies after trial. Developers of E-Mental health interventions should be aware of this need. E-Mental Health projects should plan to develop specific information material to achieve successful implementation at the end of a trial. On the other hand, intervention-related issues such as privacy concerns, doubts about the effectiveness of the program, or anonymity of the internet were rarely stated as a barrier. Likewise, patient-related issues such as negative feedback or lack of interest from patients did not seem to play a major role.“  (line 307 - line 320, tracked changes modus)

b) necessary implementation measures for the majority of the reagents almost 50% say yes and the other 50% say no:

„Still, measures need to be taken to explain the benefits and advantages of e-mental health interventions and to strengthen knowledge about their use in patient care. The present study makes some suggestions with regard to GPs. The heterogeneity of results in the spin-off survey points toward the application of broadly diversified measures, since one measure may not fit all. Accordingly, the present study shows that the majority of GPs favors certain implementation aids (e.g. information material 96.9%, application videos 80.0%). Consequently, future implementation support should include appealing informational materials and explainer videos. Other implementation aids are favored less often, but still by a considerable amount of GPs (e.g. online training seminar 50.0%). The results suggest that there might be the need to offer a range of implementation supports, leaving it to the health care professional to choose an appropriate measure.“ (line 347 - line 358, tracked changes modus)

c) possibilities to increase the level of awareness, on all of the reagents the majority say yes: 

„In the present study, the majority of GPs affirmed the listed strategies to increase the level of awareness. The dissemination of information should be considered via work-shops, conferences and professional journals. In addition, information via treatment guidelines and professional societies seem to play an important role. Another strategy could involve the distribution of information material via Internet, by postal mail or via health insurance companies. Contacting via social media was less appealing for GPs.“ (line 359 - line 364, tracked changes modus)

Round 2

Reviewer 2 Report

Thank you for your revisions. Significant improvements are evident and comprehension and value has greatly increased. Thank you.